# Immunopathogenesis of Systemic Lupus Erythematosus: Interplay of Innate and Adaptive Immunity, Microbiome Dysbiosis, and Emerging Therapeutic Targets

**DOI:** 10.3390/pathophysiology32040061

**Published:** 2025-11-10

**Authors:** Arslan Ahmed, Siru Li, Jane J. Yu, Wen-Hai Shao

**Affiliations:** 1Department of Internal Medicine, Mayo Hospital Lahore, King Edward Medical University, Anarkali Bazaar, Lahore 54000, Pakistan; drarslanahmed2@gmail.com; 2Division of Pulmonary, Critical Care and Sleep Medicine, Department of Internal Medicine, College of Medicine, University of Cincinnati, Cincinnati, OH 45267, USA; li3s7@mail.uc.edu (S.L.); yuj9@ucmail.uc.edu (J.J.Y.); 3Department of Microbiology and Immunology, AT Still University, Kirksville, MO 63501, USA

**Keywords:** apoptotic cell clearance, IFN-I, immune complex, inflammatory cytokine, pathophysiology, systemic lupus erythematosus

## Abstract

Systemic lupus erythematosus is a multifactorial autoimmune disease characterized by the dysregulation of both innate and adaptive immunity, resulting in chronic inflammation, autoantibody production, and multi-organ damage. Innate immune dysfunction involves macrophages, neutrophils, plasmacytoid dendritic cells, natural killer cells, and the complement system, which collectively amplify autoimmunity through defective clearance of apoptotic cells, overproduction of pro-inflammatory cytokines, and abnormal type I interferon signaling. Adaptive immune abnormalities, including skewed T-cell subsets, impaired regulatory T and B cells, and autoreactive B-cell hyperactivity, further perpetuate pathogenic autoantibody generation. Gut microbiota dysbiosis contributes to SLE pathogenesis via Th17 activation, loss of mucosal tolerance, and molecular mimicry mechanisms. This review synthesizes current knowledge on the immunopathogenesis of SLE, emphasizing the interplay between innate and adaptive immunity and integrating evidence from both human and experimental murine models to provide a comprehensive understanding of disease mechanisms.

## 1. Introduction

Systemic lupus erythematosus (SLE) is a recurrent autoimmune disease with flares and remissions, with the potential to cause severe damage to multiple organ systems, including kidneys, central nervous system, joints, and skin. It is characterized by the production of autoantibodies and formation of immune complexes (ICs) that deposit in tissues, leading to inflammation and organ injury [1]. SLE pathogenesis is multifactorial, involving impaired clearance of apoptotic cells, excessive activation of innate and adaptive immune responses, complement activation, and tissue inflammation. Over time, these overlapping mechanisms create a self-sustaining autoimmune feedback loop, resulting in highly heterogeneous clinical presentations across patients [2].

Type I interferons (IFN-I), particularly IFN-α, play a prominent role in transcriptional regulation of pro-inflammatory genes, an effect known as the IFN-signature [3]. The main antigens that are attacked by SLE autoantibodies are nuclear components; these are double-stranded DNA (dsDNA), chromatin, small nuclear ribonucleoproteins (snRNPs), and phospholipids [4]. Existing therapies are based on non-specific immunosuppressants, which result in serious side effects and failed to induce remission in approximately 20–30% of patients. Nearly one-third of patients in remission have a relapse following initial response [5]. Murine models of SLE have contributed significantly to our understanding of SLE pathophysiology [6].

We conducted this narrative review in accordance with the Scale for the Assessment of Narrative Review Articles (SANRA) criteria [7]. A comprehensive literature search was performed using PubMed/MEDLINE and Google Scholar databases up to January 2025, employing key terms including “Systemic Lupus Erythematosus,” “Autoimmunity,” “Innate Immunity,” “Adaptive Immunity,” “Cytokines,” “Gut Microbiota,” and “Epigenetics”. Preference was given to pathology and mechanistic investigations relevant to SLE immunopathogenesis and translational research.

This review provides a summary of recent advances in the pathophysiology of SLE, including the immune dysregulation, cytokine networks, and IFN pathways. We integrated both clinical and experimental data in this review, with most studies derived from human and a few from murine models, providing a comprehensive translational perspective.

## 2. Epidemiology

SLE prevalence varies widely, ranging from 3.7 to 49.0 per 100,000 in the US Medicare population [8]. Direct regional comparisons are challenging due to disparities in healthcare access, inconsistent case ascertainment, methodological differences, and true variation related to population structure, genetics, socioeconomics, and environmental factors [9]. Prevalence estimates range from 48 to 366.6 per 100,000 in North America, 29.3–210 in Europe, 24.3–126.3 in South America, 20.6–103 in Asia, 13–52 in Australasia, and 601.3–7713.5 in Africa [8]. The wide variation in prevalence may reflect differences in genetics, environmental exposures, healthcare access, and methodological approaches, as well as potential underdiagnosis in certain regions.

## 3. Gender-Based Variations

Women are mostly affected by SLE with the ratio of female to male varying throughout the population ranges of 1.2:1 to 15:1 with the majority of studies indicating an average of 9:1 [9,10]. These sex-specific differences are multifactorial, involving genetic, hormonal, and epigenetic influences that shape immune system variations between men and women. Testosterone has been shown to inhibit dsDNA antibody production and, to a lesser degree, overall antibody production by B cells [11]. In contrast, women demonstrate higher frequencies of anti-Ro/SSA autoantibodies, aligning with the increased prevalence of secondary Sjögren’s syndrome [12]. Estrogen likely contributes by increasing plasma cell production and ultimately autoantibody formation [13]. These hormonal influences interact with other genetic and environmental factors, collectively shaping disease onset and severity in women.

## 4. Environmental Predisposition to SLE

### 4.1. Hormonal Involvement

Estrogens are considered immune-stimulatory hormones. Beyond reproductive tissues, estrogen receptors (ERα and ERβ) are widely present in immune cells, affecting both innate and adaptive responses [13]. ERα is the predominant isoform in immune cells and has been linked to induction of IFN-γ, impairment of immune tolerance, and increased production of pathogenic autoantibodies [14]. In contrast, ERβ often counterbalances ERα activity, suppressing its transcriptional effects and exerting modest immunosuppressive influence [15]. One mechanism by which estradiol enhances autoimmunity is through ERα-mediated downregulation of the Autoimmune Regulator (AIRE) gene, a central element of thymic tolerance. AIRE expression is also hormonally regulated, decreased by progesterone and enhanced by dihydrotestosterone [15]. These findings suggest that ERα signaling broadly promotes immune activation, whereas ERβ contributes to partial restraint.

Estradiol, the most potent form of estrogen during reproductive years, has further been shown to increase markers of T-cell activation such as calcineurin and CD154 in SLE patients but not in healthy controls, reflecting heightened disease-specific sensitivity. Additionally, estradiol’s regulation of calreticulin is disrupted in SLE T cells, leading to abnormal signaling [16]. By contrast, androgens exert broadly immunosuppressive effects. Testosterone suppresses B-cell differentiation in the bone marrow; male androgen receptor–knockout mice display elevated B-cell precursors, and women with active SLE often exhibit reduced plasma testosterone and androstenedione levels [17]. Overall sex hormone contribution in the pathogenesis of SLE is summarized in Figure 1.

### 4.2. Effect of UV Radiations

Ultraviolet radiation (UVR) is another key environmental trigger. UV exposure induces pro-inflammatory cytokines, including IFN-α, interleukin-1 (IL-1), IL-6, and tumor necrosis factor-alpha (TNF-α), promoting early activation of immunity and contributing to skin lesions in SLE [18]. UVR also causes oxidative DNA modifications via neutrophil extracellular traps (NETs), leading to IFN-I production [18]. Additionally, UVR upregulates adhesion molecules (intercellular adhesion molecule-1 (ICAM-1) lymphocyte function-associated antigen 1 (LFA-1) and chemokines (IL-8, CCL5, CCL20, CCL22, and chemerin), which recruit immune cells to inflamed tissues. Chemerin specifically attracts plasmacytoid dendritic cells (pDCs) via ChemR23, a critical mechanism in SLE pathogenesis [18]. UVB exposure further induces DNA hypomethylation in CD4^+^ T cells, promoting autoreactivity and autoantibody production [19]. These UV-induced epigenetic changes illustrate how environmental insults can directly alter immune regulation at the molecular level, contributing to SLE pathogenesis (Figure 1).

### 4.3. Viral Infections

Viruses interact in multiple ways with the host immune system. These mechanisms include molecular mimicry, in which viral antigens structurally resemble self-antigens and activate autoreactive T lymphocytes; epitope spreading, where persistent viral infections trigger autoantibodies against multiple epitopes, broadening the immune response; and superantigen production, in which superantigens bind to T-cell receptors (TCRs) and major histocompatibility complex (MHC) class II molecules non-specifically in absence of antigen peptide binding, activating a wide range of T lymphocytes [20,21,22,23] (summarized in Figure 1). Bystander activation occurs when cytokines released by antigen-presenting cells (APCs) or virus-specific T cells activate neighboring autoreactive T cells [24], while altered apoptosis and clearance deficits increase cell apoptosis and release nuclear material, with impaired clearance promoting autoreactive B cell survival [25]. Viruses can also modulate immune-related gene expression through epigenetic mechanisms, including DNA methylation, histone modifications, and RNA-based regulation [26]. Persistent or recurrent infections stimulate polyclonal lymphocyte expansion and autoantibody production, and viral DNA or RNA can activate innate immunity through pattern recognition receptors (PRRs) [27]. Viral nucleic acids (NAs), along with other pathogen- or damage-associated molecular patterns (PAMPs or DAMPs), are recognized by several PRRs, including Toll-like receptors (TLRs), nucleotide-binding and oligomerization domain receptors (NLRs), RIG-I-like receptors (RLRs) such as retinoic acid-inducible gene I (RIG-I) and melanoma differentiation-associated gene 5 (MDA-5), as well as cyclic GMP-AMP synthase (cGAS) [27]. Upon sensing viral NAs, these receptor complexes may be packaged into apoptotic blebs to stimulate DCs and B cells, or trigger diverse intracellular signaling pathways that ultimately lead to the induction of proinflammatory cytokines and IFN-I response genes, particularly in pDCs [28].

Viruses involved in the pathogenesis are Epstein–Barr virus (EBV), parvovirus B19 (B19V), retroviruses (RVs), and cytomegalovirus (CMV), with EBV presenting with the strongest association. EBV persists latently in memory B cells of approximately 95% of the global population and can reactivate periodically. Primary infection occurs via the pharyngeal epithelium, followed by infection of resting B cells, T cells, NK cells, and neutrophils [29]. Increased apoptosis of EBV-infected cells can trigger both innate and adaptive immunity to cellular and viral antigens, and in genetically susceptible individuals, inadequate control of EBV may contribute to SLE development [30]. Molecular mimicry is a key mechanism in this context: antibodies generated against EBV nuclear antigen-1 (EBNA-1) can cross-react with SLE-associated autoantigens. This response, combined with epitope spreading, can ultimately lead to autoantibody production and the onset of SLE [30]. Such viral triggers exemplify the complex interplay between persistent infections and host immune dysregulation in genetically predisposed individuals.

### 4.4. Drug Induced Lupus (DIL)

DIL-causing drugs include hydralazine and procainamide (high risk), isoniazid (moderate risk), minocycline, and TNF-α inhibitors (very low risk) [31]. Procainamide, hydralazine, and isoniazid share aromatic amine or hydrazine structures and are primarily metabolized by N-acetyltransferase (NAT) enzymes, making individuals who are slow acetylators more susceptible to autoantibody accumulation following exposure [32]. Genetic predispositions, such as human leukocyte antigen (HLA)-DR4, HLA-DR0301, and complement C4 null alleles, have also been implicated, with some variation depending on the specific drug [33]. At the cellular level, hydralazine and procainamide can induce DNA hypomethylation in T cells, with procainamide acting as a competitive DNA methyltransferase inhibitor and hydralazine blocking extracellular signal-regulated kinase (ERK)-mediated induction of the enzyme. This epigenetic alteration leads to increased LFA-1 expression and enhanced autoreactivity [32,33]. The mechanisms underlying TNF-α inhibitor–induced lupus are less well understood but may involve IL-10 upregulation, B-cell hyperactivity, T-helper 2 (Th2)–driven B-cell activation or reduced cytotoxic T-cell apoptosis [34]. Collectively, these findings highlight how both drug structure and host susceptibility contribute to the development of DIL. This mechanism underscores the broader principle that environmental exposures can interact with genetic and epigenetic susceptibilities to precipitate autoimmunity.

### 4.5. Alcoholism and Cigarette Smoking

Moderate alcohol consumption decreases SLE risk, as it suppresses cellular immunity to immunogens and decreases the formation of pro-inflammatory cytokines, including TNF, IL-6, and IL-8 [35]. Antioxidants present in wine and beer, such as resveratrol and humulones, may modulate cytokines like IFN-γ and inhibit enzymes involved in the synthesis of DNA [36], while consumption of alcohol in moderation can also lower both serum immunoglobulin G (IgG) levels [37] and urinary neopterin, a macrophage activation marker corelated with SLE disease activity [38].

Cigarette smoke contains numerous toxic substances, including polycyclic aromatic hydrocarbons (PAHs), which generate reactive oxygen species (ROS) that damage DNA and proteins, forming adducts that can initiate autoantibody production [39,40] (Figure 1). Both current smoking and cessation within the last 4–5 years increases the risk of anti-dsDNA SLE, especially in individuals with a history of more than 10 pack-years [41,42]. Mechanistically, cigarette smoke promotes immune dysregulation through several pathways. It upregulates Fas (also known as CD95) on B cells and CD4^+^ T cells, heightening their sensitivity to apoptosis; excessive cell death can overwhelm the clearance of apoptotic material, potentially inducing autoimmune responses [40,41]. Smoking also impairs T-cell and NK cell function and disrupts both humoral and cell-mediated immunity. Furthermore, smoke-derived benzopyrenes activate the aryl hydrocarbon receptor, modulating Th17 and Th22 cell activity, which contributes to inflammatory and autoimmune processes in SLE [42,43]. Collectively, these effects illustrate how cigarette smoke acts as an environmental trigger that interacts with genetic and immunologic factors to promote SLE pathogenesis. Similarly, other lifestyle factors, such as moderate alcohol intake, may modulate immune responses in a protective or deleterious manner, depending on context and dose.

## 5. Genetic Predisposition to SLE

Genetic and epigenetic factors play a complex role in SLE, influencing immune regulation, activation of lymphocytes, and the clearance of apoptotic material. The MHC on chromosome 6p21.3 is consistently identified as the strongest genetic risk region for SLE [44,45]. Around 200 genes are present in this region, including nine classical HLA genes—three Class I (HLA-A, -B, -C) and three Class II pairs (HLA-DP, -DQ, -DR). Among these, HLA-DRB115:01, often inherited with HLA-DQB106:02 as part of the extended DR15 haplotype, shows the most robust association with SLE [46]. Another important risk haplotype, HLA-DRB1*03:01 (DR3), is closely linked with deletion of complement component C4 in the MHC class III region. Complement C4 is essential for clearing ICs, and its deficiency leads to accumulation of apoptotic debris, enhanced antigen presentation, and activation of autoreactive lymphocytes, central to SLE pathogenesis [47,48]. While population studies reveal differences in haplotype frequencies and linkage disequilibrium (LD), such as weaker LD in African Americans [47], C4A deletion remains a major driver of susceptibility, with effects more pronounced in males, likely reflecting their higher baseline complement levels [49]. These observations highlight how inherited genetic variations set the stage for downstream immune dysregulation.

In addition to classical HLA associations, genome-wide association studies (GWAS) have found over 170 non-HLA loci that contribute to SLE risk, including recent discoveries in East Asian populations [50]. Collectively, these loci highlight key pathogenic pathways: dysregulated IFN-I signaling (IRF3, IRF5, STAT4, IRAK1), aberrant lymphocyte activation (PTPN22, BLK, BANK1, LRRK1), impaired clearance of apoptotic cells (FCGR2A, ITGAM, NCF1), and oxidative stress [51,52]. For example, the missense variant NCF1 rs201802880 reduces ROS production, which can impair microbial killing and redox-sensitive signaling, favoring autoreactivity [53]. Together, HLA and non-HLA genetic variants create a susceptibility framework that interacts with epigenetic and environmental factors to influence SLE onset and severity.

Beyond conventional genetic polymorphisms, sex-linked genetic mechanisms contribute significantly to the higher susceptibility of women to SLE. Every woman carries two X chromosomes. Though one is normally inactivated to maintain genetic balance, certain immune-related genes escape this process. Notably, TLR-7 and its adaptor, TLR Adaptor Interacting with Endolysosomal SLC15A4 (TASL), are among the genes that escape X-chromosome inactivation (XCI), resulting in their higher expression in female immune cells. This overexpression enhances IFN signaling and immune activation, contributing to the heightened susceptibility of women to lupus [54]. Such increased expression of several X chromosome-linked genes has been proposed in lupus, allowing certain immune-related genes such as TLR-7, TASL, CD40 ligand (CD40L), and Bruton’s Tyrosine Kinase (BTK) to be expressed from both X chromosomes in female cells. This enhanced gene dosage amplifies immune signaling and supports plasmablast differentiation and IgG class switching, contributing to the stronger humoral autoimmunity observed in women with lupus [55]. In murine models, disruption of the X-chromosome inactivation machinery, particularly involving X-Inactive Specific Transcript (XIST) and its ribonucleoprotein complex, may also contribute to female-biased autoantibody generation and organ pathology [56]. Further studies show that reduced XIST expression leads to female-specific lupus-like autoimmunity, while extracellular *xist* RNA itself may stimulate TLR-7 as an endogenous ligand [57]. Together, these findings highlight disrupted X-linked gene silencing as a mechanistic basis for the pronounced female predominance in SLE.

## 6. Immune Dysregulation in SLE Pathogenesis

### 6.1. Innate Immune Dysregulation in SLE

Understanding the individual roles and interactions of innate immunity components provides insight into SLE pathogenesis. The dysregulation of these cells and pathways results in the breakdown of immune tolerance, sustained inflammation, and multi-organ involvement that characterizes SLE (Figure 2).

Macrophages serve as central effectors of innate immunity by recognizing, phagocytosing, and degrading pathogens and apoptotic cells, while also acting as APCs to activate adaptive immune responses [58]. Macrophages can polarize into pro-inflammatory M1 (classically activated) or anti-inflammatory M2 (alternatively activated) phenotypes depending on microenvironmental cues [59]. Th1 cytokines, such as IFN-γ, or microbial components like LPS, drive M1 polarization, whereas Th2 cytokines, including IL-4, favor M2 differentiation [60]. In SLE, macrophage function is skewed toward M1 polarization, promoted by ICs, microparticles (MPs), and the nuclear alarmin high mobility group box 1 (HMGB1) [61]. HMGB1 is released by necrotic cells and acts as a DAMP, binding TLRs and the receptor for advanced glycation end products (RAGE), which activates NF-κB signaling and IFN-I production [62]. This positions macrophages not only as first-line defenders but also as central amplifiers of autoimmune inflammation. Elevated HMGB1 levels in SLE serum correlate with IFN-I levels, linking activation of innate immune to systemic inflammation by TLR-9 and RAGE on pDCs [62]. Macrophages are also influenced by genetics: TNFAIP3-interacting protein 1 (TNIP1) variants enhance TLR-7-driven IFN-I production in DCs and macrophages, further driving M1 polarization and inflammation [63]. IFN-I promotes monocyte recruitment, upregulates MHC-II and co-stimulatory molecules (CD40, CD80, CD86), and facilitates T-cell activation, amplifying autoreactive responses [64,65]. A hallmark of SLE is defective macrophage-mediated clearance of apoptotic cells, which allows nuclear autoantigens to persist and stimulate autoantibody production, perpetuating IC formation and tissue injury [66]. Experimental models demonstrate that promoting M2 polarization reduces disease severity, whereas enrichment of M1 macrophages accelerates autoimmunity [67,68].

In SLE, neutrophil function is profoundly dysregulated, producing disease-inducing cytokines such as IL-1β and B-lymphocyte stimulator (BLyS), while also serving as a major source of autoantigens through the process of NET formation [69]. One of the central mechanisms is NETosis, the process by which neutrophils release NETs that are made up of decondensed chromatin decorated with cytosolic and granule proteins [70,71]. While NETs play a protective role against pathogens, their dysregulated formation contributes to autoimmunity in SLE. These NETs normally help trap and neutralize pathogens, but in SLE, defective clearance leads to prolonged exposure of nuclear and cytoplasmic autoantigens. Then, follicular dendritic cells (FDCs) present these persistent antigens in germinal centers (GCs) to autoreactive B cells, breaking self-tolerance and driving the formation of pathogenic autoantibodies [72]. NET formation also amplifies inflammation through multiple mechanisms. ICs formed by autoantibodies and nuclear antigens can recruit additional phagocytes and stimulate the secretion of cytokines that increases the inflammation, creating a self-perpetuating inflammatory feedback loop [72]. Moreover, NETs directly stimulate macrophages and other immune cells to secrete IL-1 and IL-18 via NLR family pyrin domain containing 3 (NLRP3) inflammasome activation or Purinergic receptor P2X ligand-gated ion channel 7 (P2X7) receptor signaling, further intensifying tissue inflammation [73]. These findings illustrate the synergistic interplay between neutrophils and macrophages in sustaining chronic inflammation. Clinically, neutrophils from anti-dsDNA–positive SLE patients exhibit heightened susceptibility to NETosis compared with those from anti-dsDNA–negative patients [74].

Heme oxygenase-1 (HO-1) is a cytoprotective enzyme catalyzing heme degradation and exerts anti-inflammatory effects. Pro-inflammatory cytokines such as TNF-α and IL-1α can increase HO-1 in endothelial cells and monocytes, helping restore immune homeostasis [75]. In SLE, particularly in lupus nephritis (LN), circulating monocytes exhibit reduced HO-1 expression, which correlates with increased phagocytosis and ROS production [76]. Pharmacologic induction of HO-1 using cobalt protoporphyrin (CoPP) normalizes these functions, lowering ROS and reducing tissue injury [75]. In lupus-prone mice, HO-1 induction via hemin attenuates proteinuria, glomerular IC deposition, inducible nitric oxide synthase (iNOS) expression, and autoantibody levels, indicating its regulatory role in controlling inflammation and tissue injury in lupus [77].

The complement system is a critical effector of innate immunity, comprising over 30 proteins that mediate opsonization, cytokine production, antibody responses, and clearance of ICs and apoptotic debris [78]. Complement activation occurs through three pathways: classical, lectin, and alternative. In SLE, the classical pathway, initiated by ICs, plays an important part in facilitating inflammation and tissue injury [79]. C1q functions as a potent opsonin for apoptotic cells, activating the classical pathway and facilitating phagocytosis that prevents the accumulation of autoantigens that could trigger autoimmunity [80]. Mannose-binding lectin similarly recognizes apoptotic cells through the lectin pathway [81]. Homozygous deficiencies in early complement components, including C1q, C1r, C1s, C4, and C2, markedly increase susceptibility to SLE or SLE-like disease, underscoring the protective role of complement in maintaining self-tolerance [82]. Overall, complement components serve as a linkage between innate defense and the prevention of autoimmunity. Complement fragments C3a and C5a promote chemotaxis, vascular permeability, and ROS release, recruiting macrophages, neutrophils, and lymphocytes to sites of tissue injury via their receptors C3aR and C5aR1 [81,83]. C5a can also signal through a second receptor, C5aR2, which may attenuate or enhance inflammation depending on context [83]. Dysregulated alternative pathway activation, as seen in factor H (fH), controls C3 activation, a core element of the complement cascade. C3 knockout in lupus-prone mice leads to hypocomplementemia, accelerated renal injury, and early mortality [84]. Terminal pathway activation and membrane attack complex (MAC) formation further contribute to glomerular and tubular injury in LN [85]. These observations highlight the dual role of complement in both protective clearance and pathogenic tissue injury in SLE.

PDCs are a distinct subset of lymphoid cells distinguished by their ability to produce large quantities of IFN-I [86]. They are activated by single- and double-stranded RNA or DNA internalized through Fc receptor gamma IIA (FcγRIIa) and recognized by TLR-7 and TLR-9. TLR-7 promotes disruption of GC tolerance and stimulates extrafollicular B-cell activity, contributing to autoimmunity [87], whereas TLR-9 has a dual role, suppressing immune activation under normal conditions but promoting anti-nuclear antibody (ANA) production in autoimmune contexts [88,89]. Activation of pDCs triggers signaling pathways involving myeloid differentiation primary response 88 (MyD88) and IL-1 receptor–associated kinase 4 (IRAK4), leading to the activation of interferon regulatory factors IRF3 and IRF7 and subsequent IFN-I production [90,91].

IFN-Is produced by pDCs link innate and adaptive immunity, interacting with monocytes, neutrophils, NK cells, and T and B lymphocytes [92]. This IFN production drives extrafollicular differentiation of B cells into plasmablasts, which secrete autoantibodies such as anti-dsDNA, perpetuating a vicious cycle of autoimmunity [93]. Additionally, IFN-I enhances recruitment and differentiation of pro-inflammatory T cells, further amplifying inflammation [89].

Epigenetic mechanisms serve as a crucial interface between genetic susceptibility and environmental triggers, influencing innate immune activation in SLE. Environmental factors such as UVR, infections, and certain medications can induce persistent alterations in DNA methylation, histone modifications, and microRNA (miRNA) expression, collectively reshaping macrophage, DC, and lymphocyte responses [54]. These changes enhance cytokine production, IFN signaling, and inflammatory feedback loops characteristic of SLE.

DNA methylation defects are among the earliest and most consistent findings. CD4^+^ T cells from SLE patients exhibit global hypomethylation, leading to overexpression of immune-activation genes and autoreactive T-cell expansion [55]. Similar patterns in monocytes and pDCs augment IFN-signature. Impaired methylation on the inactive X chromosome, particularly hypomethylation of X-linked immune genes such as CD40L, contributes to female predominance by enhancing costimulatory signaling [56]. Histone modifications, including acetylation, phosphorylation, and methylation, further regulate chromatin accessibility [57]. In SLE, global hypoacetylation of histones H3 and H4 promotes transcription of pro-inflammatory genes [94], while increased histone 3 lysine 27 trimethylation (H3K27me3) and reduced histone 3 lysine 9/14 acetylation (H3K9/K14ac) at the miR-1246 promoter suppress regulatory miRNA expression, driving B-cell and DC activation [95]. miRNAs fine-tune these processes post-transcriptionally. Upregulated miR-301a-3p enhances IL-6, IL-17, and IFN-γ secretion through the IRAK1–Pellino E3 ubiquitin protein ligase 1 (Peli1) pathway [96], whereas downregulated miR-99a-3p and miR-124 impair immune regulation by promoting B-cell autophagy and T-cell hyperactivity [97,98].

NK cells play dual roles in immunity, contributing to both protective and pathogenic responses depending on the immune context, target organ, and NK subset analyzed [99]. They influence inflammation through the production of cytokines such as granulocyte/macrophage-colony stimulating factor (GM-CSF), macrophage-colony stimulating factor (M-CSF), IL-5, IL-10, IL-13, IFN-γ, and TNF-α, as well as chemokines including CCL3, CCL4, CCL5, IL-8, RANTES, and XCL1 [100]. NK cells may provide protection against SLE by eliminating DCs; however, peripheral NK cell cytotoxicity is impaired in SLE patients regardless of disease activity [101]. NK cells also contribute to SLE pathogenesis via interactions with pDCs. They enhance IFN-α production by IC-activated pDCs through CCL4 secretion and LFA-1/DNAM-1–dependent contact, while pDC-derived IFN-α promotes NK cell development, maturation, and IFN-γ production [99]. This bidirectional activation amplifies inflammation, creating a cytokine- and chemokine-rich environment that includes IFN-α, IFN-γ, TNF-α, IL-6, IL-8, CCL3, and CCL4 [90].

An atypical NK/pDC subset identified in murine lupus models, expressing NK1.1, CD11c, CD122, and MHC-II responds to IL-15, produces type I and II IFNs, exhibits high proliferation, and demonstrates both cytotoxic and antigen presentation. Adoptive transfer of these cells induces lupus-like autoimmunity [102]. This highlights the importance of distinct NK/pDC subsets in disease initiation and progression, emphasizing potential cellular targets for therapy.

The pervasive IFN-signature observed in SLE represents a mechanistic bridge between innate and adaptive immunity. Building upon the epigenetically primed state of innate immune cells, persistent activation of pDCs and monocytes drives sustained IFN-I release, which in turn enhances antigen presentation and promotes extrafollicular B-cell differentiation [87,93]. IFN-I upregulates TLR-7 and TLR-9 pathways in B cells, increases expression of B-cell activating factor (BAFF) and A Proliferation-Inducing Ligand (APRIL), and supports the emergence of age-associated B cells that secrete high-affinity autoantibodies [87,103,104]. This cytokine-driven extrafollicular activation bypasses GC regulation, leading to rapid expansion of autoreactive plasmablasts and diversification of the autoantibody repertoire [57,104]. Together, these IFN-mediated processes exemplify how dysregulated innate immune signaling orchestrates downstream adaptive immune responses, perpetuating the cycle of systemic autoimmunity in SLE.

### 6.2. Adaptive Immune System

#### 6.2.1. T Cells

CD4^+^ T cells interact with autoreactive B cells, providing essential signals that drive their activation and differentiation into antibody-secreting cells, and promoting class switching from low-affinity, polyreactive IgM to high-affinity IgG autoantibodies [105]. This T-cell crosstalk is pivotal for the generation of pathogenic autoantibodies in SLE. In addition, CD4^+^ T cells in SLE show a marked imbalance between effector and regulatory subsets, with a dominance of pro-inflammatory Th1, Th17, and Tfh cells, which contribute to disease exacerbation [105,106] (Figure 2).

Th1 cells release IL-2 and IFN-γ, which support inflammation, B-cell class switching, GC formation, and autoantibody production [99]. Th2 cells secrete IL-4, IL-5, and IL-13, aiding humoral immune responses. In lupus-prone mice, IL-4 levels are elevated and can drive IgG-to-IgE class switching, though human studies show more variable results [107]. Altered Th1/Th2 balance in SLE contributes to heightened autoantibody responses and activity of the disease. In SLE patients, decreased IL-4-forming Th2 cells and an elevated IFN-γ/IL-4 ratio are correlated with higher activity of the disease. Additionally, IL-13 and IL-5 in certain patient subsets contribute to B-cell differentiation and antibody production [108].

Th17 cells mainly secrete IL-17, recruiting neutrophils, activating innate immunity, and enhancing B-cell function. They act synergistically with BAFF to promote B-cell proliferation and antibody formation [105,109]. IL-23, essential for Th17 maintenance, is elevated in SLE, particularly in patients with renal involvement, and increased levels are predictive of poor treatment response [110,111]. Th22 cells, which mainly produce IL-22, may have either protective or pathogenic roles depending on the microenvironment context, treatment, or disease stage [105,112].

Th9 cells, regulated by IRF4, secrete IL-9, helping B-cell proliferation and autoantibody formation in lupus-prone mice. IL-9 neutralization improves nephritis outcomes, though some studies suggest anti-inflammatory roles as well [113,114].

Regulatory T cells (Tregs) serve as critical regulators of the immunity, preserving self-tolerance and limiting autoimmunity. They are characterized by expression of CD4, CD25 (IL-2 receptor α-chain), and FoxP3. CD4^+^CD25^+^FoxP3^+^ Tregs inhibit activation and expansion of CD4^+^ helper T cells, cytotoxic CD8^+^ T cells, and B cells [115], exerting their suppressive function through secretion of anti-inflammatory cytokines such as IL-10 and TGF-β [116]. In SLE, Treg impairment is a critical factor enabling uncontrolled autoreactive immune responses. Both the number and function of Tregs are compromised, leading to a breakdown of immune tolerance. These abnormalities are characterized by diminished suppressive activity, reduced survival, and altered responsiveness to cytokines [117,118]. IL-2, critical for Treg development and maintenance, is reduced in SLE patients alongside CD25 expression [115]. Additionally, IFN-α can activate IL-1 receptor-associated kinase 1 (IRAK1), inducing Treg apoptosis and creating imbalance between Tregs and effector T cells [119]. Platelet interactions via P-selectin and P-selectin glycoprotein ligand-1 (PSGL-1) further impair follicular Treg (Tfr) function by triggering Syk phosphorylation and increasing intracellular calcium, leading to downregulation of the TGF-β axis and limiting immunosuppressive activity [117,120].

Tfh cells play an important part in GC reactions, contributing to B- and T-cell differentiation, proliferation, and antibody formation through IL-21 secretion [121]. Dysregulation of Tfh cells can promote pathogenic autoantibody generation and is implicated in SLE development [103]. Altered cytokine levels, including IL-12, IL-23, and TGF-β, may impair circulating follicular regulatory T cell (cTfr) function [103]. In SLE, elevated IL-21 levels reflect an increased number of circulating Tfh (cTfh) cells, which correlate with anti-dsDNA antibody titers [121,122]. Concurrently, cTfr cell numbers are reduced, potentially serving as an indicator of active disease [122]. Thus, an imbalance between Tfh and Tfr subsets perpetuates humoral autoimmunity. Aberrant Tfh expansion is also mediated by the OX40 ligand (OX40L)-OX40 axis; increased OX40L expression on APCs correlates with disease activity, and OX40 stimulation drives naive and memory CD4^+^ T cells toward Tfh differentiation. Blocking this pathway alleviates LN in murine models [123]. Additionally, programmed death-ligand 1 (PD-L1) and IL-4-producing basophils further help pathogenic Tfh accumulation in lupus [124].

CD8^+^ T cells are gaining recognition for their role in SLE pathogenesis, mediating cytotoxic effects through the release of perforin and granzymes. Studies indicate that SLE patients display impaired CD8^+^ T-cell cytolytic function, increasing susceptibility to infections and promoting autoimmune responses [125]. Defective CD8 responses contribute to both infection risk and autoimmune amplification in SLE. In particular, increase in CD8^+^CD38^+^ T cells in SLE patients is associated with reduced cytotoxic capacity, which occurs through suppression of cytotoxic gene expression via the NAD+/sirtuin1/EZH2 pathway [126].

Double-negative (DN) T cells have the αβ T-cell receptor (TCR) but lack CD4, CD8, and NK cell markers [127]. DN T cells likely originate from activated auto-reactive CD8 T cells that downregulate CD8 expression, although the exact mechanisms enabling these cells to escape activation-induced cell death remain unclear. Self-antigens released from apoptotic cells may contribute to this process. In SLE, DN T cells exhibit proinflammatory properties, infiltrate the kidneys, produce high levels of IL-17 and IFN-γ, and contribute to the generation of autoantibodies [106,127,128].

Gamma delta (γδ) T cells are involved in immune responses against infections and tumors and have been shown to play a role in autoimmunity. In SLE, γδT cells may have APC-specific markers such as HLA-DR or CD80/86, enabling them to activate T cells via antigen presentation, although their numbers decrease during disease progression [129]. The multifunctional roles of γδT cells underscore the diversity of T-cell-mediated contributions to SLE. One subset produces Th1/Th2 cytokines, including IFN-γ, TNF-α, IL-10, and IL-4, contributing to SLE pathogenesis [118]. Another subset, termed Tγδ17 cells and functionally similar to Th17 cells, promotes T-cell differentiation, increases B cell proliferation and antibody formation through IL-17 [129].

#### 6.2.2. B Cell Dysfunction and Autoreactivity

B cells, key contributors to humoral immunity, are identified by a B cell receptor (BCR) and express MHC class I and II, enabling uptake of antigen, processing, and presentation [130]. In SLE, B-cell dysregulation drives the formation of pathogenic autoantibodies and the breakdown of tolerance. Mature B cells differentiate into B1 cells, making low-affinity antibodies, and B2 cells, forming high-affinity, including autoreactive, antibodies that drive SLE. B2 cells subdivide into follicular (T–dependent) and marginal zone (T–independent) subsets. Their survival depends on BAFF and APRIL, facilitated by inflammatory cytokines such as TNF-α, IFN-γ, and IL-2 from DCs, macrophages, and neutrophils [117,118]. BAFF supports survival and maturation through the NF-κB pathway, while APRIL enhances switching of class and survival of plasma cells through NF-κB and MAPK pathways [131]. Abnormal B-cell function in SLE, including altered BCR signaling, defective tolerance, and impaired apoptosis, promotes autoreactive B-cell survival and autoantibody production [132].

During splenic B-cell maturation, transitional B cells with high intracellular IFN-β display increased BCR-driven survival and activation, contributing to more severe disease, notably in African American patients [133]. Early B-cell hyperactivation sets the stage for peripheral tolerance defects and autoimmunity. This early hyperactivation predisposes these cells to autoreactivity and sets the stage for subsequent defects in peripheral tolerance. After this transitional stage, B cells lose the ability to support the tolerogenic function of marginal zone macrophages, which typically clear apoptotic cell debris non-inflammatorily via upregulation of indoleamine 2,3-dioxygenase (IDO) [134]. IFN-I induced migratory changes in marginal zone precursor B cells exacerbate this defect, as they lose sphingosine-1-phosphate (S1P) tethering and breach the follicular exclusion barrier. Once inside the follicle, these MZ-P B cells act as potent antigen-presenting and pro-inflammatory cells, characterized by increased production of IL-6, reduced production of the regulatory cytokine IL-10, and expression of membrane lymphotoxin β (mLTβ), which stimulates FDCs and initiates GC formation [134,135]. At the GC and extra-GC stages, B cells further upregulate IL-17RA and IFN-γR, rendering them responsive to IL-17 and IFN-γ signals. IL-17 signaling prolongs GC retention by inducing regulator of G-protein signaling 13 (RGS13) and RGS16, enhances activation-induced cytidine deaminase (AICDA) expression, and strengthens interactions with Tfh cells, promoting antibody diversification [136]. Concurrently, IFN-γ signaling drives the emergence of T-bet^+^CD11c^+^CXCR5^−^IgD^−^CD27^−^ (DN2) autoreactive B cells, a highly pathogenic subset implicated in autoantibody production and SLE progression [104]. These signaling pathways explain how B cells transition from benign to pathogenic states in SLE.

Beyond developmental defects, SLE patients also show abnormalities in memory B-cell and plasma cell compartments. Plasma cells, derived from GC or memory B cells, stay in the bone marrow and spleen for longer period and produce most serum immunoglobulins, including pathogenic SLE autoantibodies [104]. Among memory B cells, CD27^+^IgD^−^ switched memory B cells are expanded, while CD27^−^IgD^−^ late memory B cells correlate with disease activity, renal involvement, and autoantibody presence [137]. Circulating plasmablasts and tissue-resident plasma cells, especially in the kidneys, sustain autoantibody production. Defective regulatory B cell (Breg) activity further exacerbates autoreactive B-cell persistence. B-cell depletion therapy may be less effective due to dysfunctional Bregs, which normally produce IL-10, IL-35, and TGF-β to decrease immune responses. In SLE, Bregs may be numerically normal or elevated but functionally impaired, producing insufficient IL-10 and failing to restrain autoreactive B cells [138,139].

The autoantibody repertoire in SLE is highly diverse, encompassing over 200 distinct specificities that can target nucleic acids, lipids, proteins, organelles, cells, plasma proteins, or tissues [140,141]. Anti-dsDNA antibodies stimulate TNF-α, IL-1β, IL-6, IL-8, and iNOS, promoting tissue injury, particularly in kidneys [136,142]. Most pathogenic autoantibodies are IgG, while IgM may help clear apoptotic debris [142]. IgE autoantibodies stimulate pDCs to secrete IFN-α, TNF-α, and IL-6, contributing to disease progression [118,143]. IgE-mediated ICs drive inflammation without causing classic allergy symptoms, and IgE blockade reduces IFN-α production and delays disease onset in animal models [143].

Despite this diversity, the core diagnostic assays rely on ANA testing, lupus anticoagulant (LA), and autoantibodies against dsDNA, Smith (Sm), U1 ribonucleoprotein (U1 RNP), Ro, and La, which remain the most established markers for disease detection [138]. High-throughput approaches have identified additional extracellular and secreted targets that may contribute to disease heterogeneity. Using rapid extracellular antigen profiling (REAP), Wang et al. discovered autoantibodies in SLE patients targeting 84 extracellular and secreted proteins, including cytokines (IL-6, INF-1, IL-1α, TNF-α), chemokines (CXCL3, CCL8, CCL22), growth factors (VEGF-B, FGF-21), and immunoregulatory proteins (CD44, CD95, PD-L2, butyrophilin-like 8 (BTNL8], B7-H4) [144]. These findings underscore the extensive yet individualized nature of humoral autoimmunity in SLE. Although each autoantibody was rare (<5%), this study highlights the potential breadth of extracellular targets, while also noting that some known antigens (β2-glycoprotein I (B2GPI), C1q, deoxyribonuclease 1 (DNASE1), apolipoprotein A1 (ApoA1)) were not detected, suggesting limitations of the technique [144].

SLE patients also exhibit anti-mitochondrial antibodies (AMAs), targeting the mitochondrial outer membrane, mitochondrial DNA (mtDNA), mtRNA, and heat shock protein 60 (Hsp60) [145]. Mechanisms such as mitochondrial stress, damage, or defective clearance likely expose these antigens, with neutrophils serving as a significant source of mitochondrial components extracellularly [146]. Recently, IgG2 antibodies against superoxide dismutase 2 (SOD2), a detoxifying enzyme in mitochondria, were reported in SLE, further implicating mitochondrial autoimmunity in disease pathogenesis [143]. Collectively, these B-cell and autoantibody abnormalities illustrate the adaptive immunity’s role in SLE pathogenesis (Figure 2).

## 7. Gut Microbiota Dysbiosis in SLE

SLE patients consistently exhibit marked alterations in gut microbial composition compared with healthy individuals, characterized by enrichment of *Proteobacteria*, *Bacteroidetes*, and *Actinobacteria*, accompanied by depletion of *Firmicutes* [147]. These compositional deviations, confirmed across multiple independent human and murine investigations [148,149,150], represent a reproducible microbial signature of lupus-associated dysbiosis. In non-active human SLE cohorts, reductions in *Firmicutes* families such as *Ruminococcaceae* and *Lachnospiraceae* have been observed, alongside elevations of *Prevotellaceae*, *Bacteroidaceae*, and *Streptococcus* species (including *S. pneumoniae* and *S. intermedius*), suggesting an oral–gut microbial axis bridging mucosal compartments [149]. Murine lupus-prone models demonstrate parallel findings, in which declining *Lactobacillus* levels and rising *Lachnospiraceae* abundance correlate with disease progression [149]. In active LN, the selective expansion of *Ruminococcus gnavus* (*RG*) correlates positively with anti-dsDNA titers and SLE disease activity but inversely with complement components C3 and C4, linking dysbiosis to IC formation and renal involvement [151].

Loss of microbial equilibrium disrupts epithelial homeostasis and increases intestinal permeability, facilitating systemic exposure to microbial components. Evidence from both human and animal models indicates enhanced barrier permeability, characterized by elevated fecal calprotectin, albumin, and zonulin, along with heightened serum levels of LPS, soluble CD14, and α_1_-acid glycoprotein [151,152]. In SLE patients, plasma zonulin correlates positively with C-reactive protein (CRP) and inversely with C3 [153], indicating subclinical barrier dysfunction even during disease quiescence. Experimental studies show that translocating pathobionts such as *Enterococcus gallinarum* can cross the intestinal mucosa, colonize secondary lymphoid tissues, and initiate Th17–dominated autoimmunity [154]. Collectively, these findings suggest that compromised mucosal integrity, often referred to as “leaky gut,” is not merely a consequence but an active amplifier of systemic immune activation in lupus.

Microbial dysbiosis also drives immune dysregulation through molecular mimicry and cross-reactivity between microbial and host antigens. *Bacteroides thetaiotaomicron* expresses Ro60-like sequences capable of inducing anti-Ro60 antibodies and epitope spreading toward Ro52, Sm, and U1-RNP antigens [155,156]. Similarly, *Ruminococcus gnavus*, *Odoribacter splanchnicus*, and *Akkermansia muciniphila* express peptides structurally homologous to lupus autoantigens, stimulating IFN-γ, IL-17, and IgG_3_ responses [154]. The detection of elevated systemic antibodies against whole bacterial lysates in human SLE, an observation not previously reported, reinforces the concept that microbial antigens may reach systemic circulation through translocation or mimicry, subsequently driving autoantibody diversification [150]. Heat-shock proteins derived from commensal organisms further potentiate cross-reactive immune responses, explaining the presence of anti-HSP autoantibodies in SLE [157]. These findings collectively illustrate how the gut microbiota may act both as an initiator and a perpetuator of systemic autoimmunity through continuous antigenic stimulation.

Parallel to antigenic mimicry, dysbiosis perturbs immune balance by modulating key regulatory cell populations. Butyrate-producing *Firmicutes*, including *Faecalibacterium prausnitzii*, *Anaerostipes*, *Coprococcus*, and members of the *Ruminococcaceae* family, generate short-chain fatty acids (SCFAs) that strengthen epithelial tight junctions and promote differentiation of retinoic acid receptor-related orphan receptor gamma t (RORγt)^+^ Tregs, maintaining mucosal tolerance [150,158]. Their depletion diminishes SCFA availability, weakens epithelial integrity, and reduces Treg-mediated suppression. Conversely, expansion of pro-inflammatory taxa such as *Prevotella*, *Bacteroides*, *Bilophila*, and *Parabacteroides* promotes Th17 differentiation and elevates systemic cytokines including IL-6, IL-17, and IFN-γ [154]. This Th17/Treg imbalance, consistently observed in both human and murine lupus models [149,152], sustains a pro-inflammatory environment conducive to IC deposition and tissue injury. Microbial metabolites further modulate host immunity: SCFAs induce histone acetylation in Tregs, while tryptophan catabolites and bile-acid derivatives regulate epithelial renewal and attenuate DC activation [159,160]. Disruption of these metabolic axes in lupus underscores the interplay between microbial ecology and systemic immune dysregulation.

Beyond immune modulation, lupus-associated dysbiosis involves profound metabolic and structural shifts. Specific genera, including *Bacteroides*, *Parabacteroides*, and *Alistipes*, correlate with glycan, protein, and adipocytokine pathways linked to SLE activity [161]. Deficiency of SCFAs such as butyrate and propionate reduces colonocyte energy supply and increases oxidative stress, while altered bile acid and tryptophan metabolism perturbs host receptor signaling and epithelial regeneration [159,160]. These interconnected disturbances amplify inflammatory signaling and metabolic stress, creating a self-reinforcing cycle of barrier dysfunction and immune activation.

Cross-species comparisons strengthen the biological validity of these findings. In human SLE, enrichment of inflammation-promoting taxa such as *Desulfovibrio piger*, *Bacteroides thetaiotaomicron*, and *Verrucomicrobia* members coincides with depletion of protective *Firmicutes* including *F. prausnitzii*, *Fusicatenibacter saccharivorans*, *Lactobacillus*, *Romboutsia*, and *Ruminococcaceae* [158]. In the pristane-induced lupus mouse model, which replicates lupus-specific serological and organ manifestations, metagenomic profiling revealed concordant alterations involving *Tenericutes*, *Tannerellaceae*, *Parabacteroides*, *Bacteroides*, and *Alistipes* [158]. These overlapping taxa delineate conserved mechanisms such as loss of butyrate producers, enhanced intestinal permeability, Th17/Treg disequilibrium, and molecular mimicry that mechanistically connect intestinal dysbiosis to systemic autoimmunity across species.

Therapeutic modulation of gut dysbiosis in SLE through diet, probiotics, prebiotics, or fecal microbiota transplantation remains an evolving field. Current evidence is preliminary, and no standardized treatment pathway has been established. Ongoing research is needed to clarify mechanisms and guide individualized therapeutic strategies.

Overall, the convergence of clinical and experimental evidence supports a mechanistic continuum wherein altered gut microbial ecology contributes to lupus initiation and propagation. Reduced microbial diversity and depletion of beneficial Firmicutes weaken mucosal tolerance and barrier integrity, whereas enrichment of Bacteroidetes and Proteobacteria amplifies inflammatory cascades and antigenic mimicry. The resulting leaky-gut environment exposes the immune system to microbial epitopes that perpetuate Th17-driven cytokine production and broad-spectrum autoantibody formation. Although current findings primarily establish strong associations and biological plausibility rather than definitive causality, their reproducibility across species underscores the gut–immune axis as a pivotal component of lupus pathogenesis.

## 8. Conclusions

SLE is driven by intricate interactions between innate and adaptive immune components, as well as environmental factors such as gut microbiota dysbiosis. Innate immune dysfunction, including aberrant macrophage polarization, neutrophil NETosis, impaired complement activity, and dysregulated pDC and NK cell functions, sets the stage for chronic inflammation and loss of self-tolerance. Adaptive immune abnormalities, particularly in T and B cell subsets, further perpetuate autoantibody production and tissue injury. Dysbiosis of the gut microbiota may amplify systemic autoimmunity through molecular mimicry, Th17 activation, and disruption of mucosal tolerance. Collectively, these mechanisms create a self-reinforcing cycle of immune dysregulation, shedding light on the complex pathways that maintain the immune imbalance and contribute to organ damage in SLE.

## 9. Future Directions

Future research should aim to unravel the intricate cross talk between innate and adaptive immune systems that drive SLE. A deeper understanding of how genetic, epigenetic, and environmental factors converge to sustain IFN signaling, defective apoptotic clearance, and loss of immune tolerance remains essential. Particular attention should be given to cell-specific mechanisms such as macrophage polarization, NET formation, pDC activation, and aberrant B and T cell differentiation that collectively perpetuate systemic inflammation. Advancing technologies such as multiomics integration, single-cell transcriptomics, and spatial immune profiling will be critical to define molecular heterogeneity among patients. Furthermore, longitudinal cohort studies and systems biology approaches can help delineate causal pathways, identify early biomarkers of disease onset or flare, and clarify sex-based immunological differences that underlie female predominance. Ultimately, such insights will refine disease classification, improve predictive accuracy, and lay the foundation for precision diagnostics and prevention strategies in SLE.

## Figures and Tables

**Figure 1 pathophysiology-32-00061-f001:**
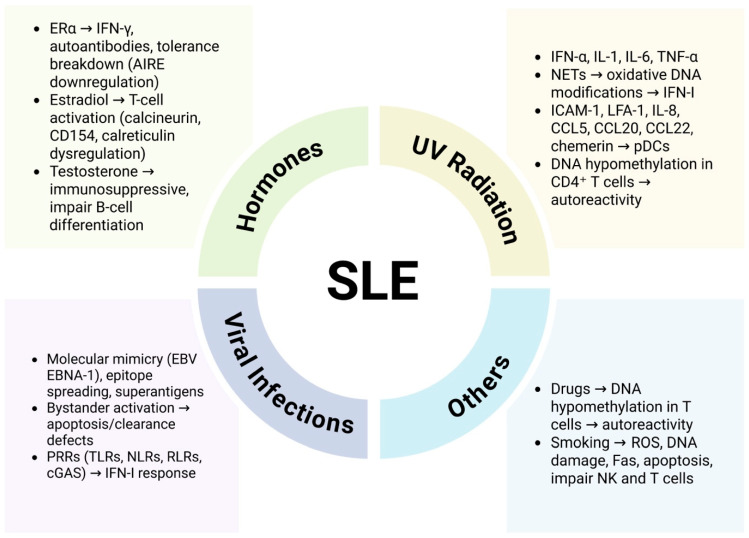
Environmental factors involved in the pathogenesis of SLE. The etiology of SLE is multifactorial and involves environmental, hormonal, and immunological triggers. Hormonal influences include ERα-driven IFN-γ production, estradiol-mediated T-cell activation, and testosterone-mediated immunosuppression. UV radiation promotes inflammatory cytokine release, oxidative DNA damage through NETs, upregulation of adhesion molecules and chemokines, and DNA hypomethylation in CD4^+^ T cells. Viral infections, particularly EBV, contribute via molecular mimicry, epitope spreading, superantigen effects, and PRR-mediated IFN-I responses. Other environmental factors, such as drug-induced DNA hypomethylation and cigarette smoking, generate ROS, trigger Fas-mediated apoptosis, and impair NK and T-cell function. Together, these pathways disrupt immune tolerance, enhance autoreactivity, and drive chronic inflammation in SLE. AIRE, autoimmune regulator; CD, cluster of differentiation; CCL, C-C motif chemokine ligand; cGAS, cyclic GMP-AMP synthase; EBNA-1, Epstein–Barr nuclear antigen 1; EBV, Epstein–Barr virus; ERα, estrogen receptor alpha; ICAM-1, intercellular adhesion molecule 1; IFN-I, type I interferon; LFA-1, lymphocyte function-associated antigen 1; NETs, neutrophil extracellular traps; NK, natural killer; NLRs, NOD-like receptors; pDCs, plasmacytoid dendritic cells; PRRs, pattern recognition receptors; RLRs, RIG-I-like receptors; ROS, reactive oxygen species; TLRs, Toll-like receptors; TNF-α, tumor necrosis factor alpha, UV, ultraviolet. Figure created with the help of BioRender.com.

**Figure 2 pathophysiology-32-00061-f002:**
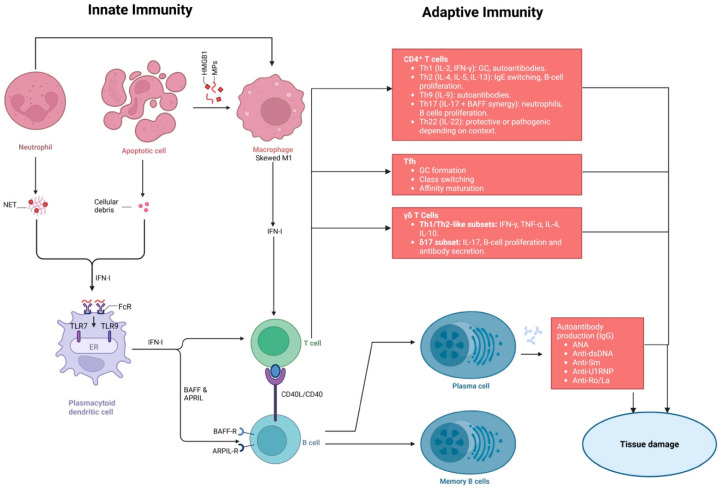
Role of innate and adaptive immunity in the immunopathogenesis of SLE. Innate immune dysregulation, including NET formation, defective clearance of apoptotic debris, and skewed macrophage M1 activation, promotes persistent IFN-I production. pDCs, activated via TLR-7, TLR-9, and Fc receptors, further amplify IFN-I signaling, enhancing T-cell and B-cell activation. In the adaptive immune system, CD4^+^ T-cell subsets (Th1, Th2, Th9, Th17, Th22) and Tfh cells contribute to GC reactions, class switching, and autoreactive B-cell proliferation. γδ T-cell subsets (Th1/Th2-like and IL-17–producing) also promote B-cell expansion. B cells, stimulated via CD40L/CD40 and cytokines such as BAFF and APRIL, differentiate into plasma cells and memory B cells, leading to pathogenic autoantibody production (e.g., ANA, anti-dsDNA, anti-Sm, anti-U1RNP, anti-Ro/La). These autoantibodies drive immune complex formation, chronic inflammation, and tissue damage characteristic of SLE. ANA, antinuclear antibody; APRIL, a proliferation-inducing ligand; BAFF, B cell-activating factor; BAFF-R, BAFF receptor; CD, cluster of differentiation; dsDNA, double-stranded DNA; ER, endoplasmic reticulum; GC, germinal center; IFN-I, type I interferon; IL, interleukin; NET, neutrophil extracellular trap; pDCs, plasmacytoid dendritic cells; Tfh, T follicular helper cell; Th, T helper cell; TLR, Toll-like receptor; TNF-α, tumor necrosis factor alpha; U1RNP, U1 ribonucleoprotein. Figure created with the help of BioRender.com.

## Data Availability

No new data were created or analyzed in this study. Data sharing is not applicable to this article.

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
