# Peer review of "Immunopathogenesis of Systemic Lupus Erythematosus: Interplay of Innate and Adaptive Immunity, Microbiome Dysbiosis, and Emerging Therapeutic Targets"

_pathophysiology, 2025, doi:10.3390/pathophysiology32040061_

Round 1

Reviewer 1 Report

Comments and Suggestions for Authors

The manuscript proposed by Ahmed et al., which presents a proposal for a review of the immunopathology of SLE, provides relevant and up-to-date data on the advances made in this field. To improve understanding of the content of the document, I make the following suggestions:

1. Verify the abbreviations used in the manuscript so that the definition of each one is respected.

2. To provide a more comprehensive view of the topic, I suggest including an additional figure that the authors consider relevant and that includes explicit data on intestinal dysbiosis.

3. To enhance the readability and understanding of the content, I recommend increasing the font size in Figure 2.

4. To provide a more detailed understanding of the topic, I suggest expanding the information included in the section on intestinal dysbiosis and including the following references: https://doi.org/10.1016/j.autrev.2025.103921; https://doi.org/10.1186/s13099-025-00683-7;  https://doi.org/10.3389/fmicb.2024.1319654

Author Response

Comment 1: Verify abbreviations throughout the manuscript so that each is defined at first mention.
Response: All abbreviations have been carefully reviewed and defined upon first appearance across the text and figure legends.

Comment 2: To provide a more comprehensive view of the topic, I suggest including an additional figure that the authors consider relevant and includes explicit data on intestinal dysbiosis.
Response: We thank the reviewer’s suggestion. There are many review articles summarized intestinal dysbiosis and inflammation (Biomedicine & Pharmacotherapy, 2023; Journal of Translational Medicine, 2022, et al). We expanded the gut microbial dysbiosis to provide more support.

Comment 2: To enhance the readability and understanding of the content. I recommend increasing the font size in Figure 2.

Response: We tried several times but encountered with the software problem. We will modify and provide with larger font size in the final proofreading version.

Comment 4: Expand the intestinal dysbiosis section and include the suggested references.
Response: The section on gut microbiota has been substantially expanded. Recommended references are also included.

Reviewer 2 Report

Comments and Suggestions for Authors

The manuscript provides a comprehensive and well-structured overview of the immunopathogenic mechanisms of systemic lupus erythematosus, clearly integrating the abnormal activation of both innate and adaptive immunity, while highlighting the role of intestinal dysbiosis as a modulatory factor. The narrative is fluid and accessible, and the two summary figures allow the reader to visualize in a synthetic manner the complex network of cellular and molecular interactions involved in the disease. The work is perceived as a solid review with the potential to serve as a reference, although in certain sections greater depth and updated content could further increase its impact and translational usefulness.

Among its main strengths are the way in which the interferon signature is linked to T and B cell activation, the description of relevant cellular subpopulations such as Tfh and DN2 B cells, and the inclusion of a specific section on the intestinal microbiota and molecular mimicry, which adds value compared to more conventional reviews. The selection of foundational references is well accomplished and lends robustness to the narrative, while the effort at graphic synthesis through the two integrative figures constitutes a very valuable resource for both researchers and clinicians.

Nevertheless, the manuscript presents some limitations that should be addressed.

  1. A brief methodological note is missing that would specify the databases consulted, the time window of the search, and the criteria for selecting the bibliography, even in the context of a narrative review. This would help define the scope of the work and allow the reader to better assess its comprehensiveness. I am not certain whether the journal prefers this style of presentation or the more narrative approach used by the authors, so I suggest this point as a consideration for the authors and/or the editor to indicate their preference.
  2. The section devoted to the microbiota is of great interest but could be enriched with some additional nuances. It would be useful to introduce the concept of intestinal permeability and its relationship with markers such as zonulin, emphasizing the potential pharmacological reversal of this phenomenon. The role of bacteria capable of translocating from the intestine to other organs, such as Enterococcus gallinarum, and their ability to trigger autoimmune responses, should also be mentioned. Likewise, a brief discussion of the relevance of microbial metabolites, such as short-chain fatty acids or tryptophan-derived pathways activating host receptors, would provide a more complete view of how dysbiosis can modulate autoimmunity. In the same context, it would be highly advisable for the authors to explicitly mention therapeutic approaches aimed at restoring microbial balance, such as the use of probiotics with immunomodulatory potential or fecal microbiota transplantation. Although still at preliminary stages, these represent a highly promising field and a potential avenue for future interventions.
  3. The section on emerging therapies warrants an update to reflect the most recent advances. In addition to investigational drugs, the authors should include a subsection distinguishing between already approved therapies and those in development. This includes the relevance of anifrolumab as an IFNAR1 antagonist, belimumab in lupus nephritis, voclosporin as a next-generation calcineurin inhibitor, and the positive results of obinutuzumab in the same indication. This organization, perhaps through a table or comparative box, would help readers position both current and future evidence in the therapeutic context of lupus.
  4. Another aspect that would enhance the text is the inclusion of sex-gender considerations and their relationship to the higher susceptibility of women to lupus, in particular the role of the X chromosome and TLR7 overexpression as mechanisms explaining the female bias. Similarly, the narrative on molecular mimicry could be strengthened by integrating not only the case of Ruminococcus gnavus and Ro60, already discussed, but also the possibility of other autoantigens involved in cross-epitope processes.
  5. Consider adding a transition paragraph underlining how interferon signatures promote extrafollicular B-cell activation and drive the expansion of the autoantibody repertoire, thereby providing a conceptual bridge between innate and adaptive immunity.
  6. It would also be advisable to close the microbiota section with a sentence clearly distinguishing evidence of association, biological plausibility, and intervention outcomes, avoiding overgeneralizations that exceed the current data.

In addition to these substantive changes, several minor editorial adjustments would be desirable:

  1. Standardize terminology by correcting expressions such as “INF-1” to “IFN-I” and “shading lights” to “shedding light.”
  2. Consistently define all abbreviations at their first mention and review word breaks and grammatical concordance. For example, in the section on B cells, the term Tfh (T follicular helper) is used directly without providing its full name at first mention. A similar issue occurs with DN2 B cells and APRIL (A Proliferation-Inducing Ligand), which are introduced only as acronyms without prior definition.

Author Response

Comment 1: Add a brief methodological note specifying databases, timeframe, and selection criteria.
Response: Addressed. A methodological note has been added after the Introduction describing the databases (PubMed/MEDLINE, Google Scholar), timeframe (up to January 2025), key search terms, and inclusion criteria per SANRA guidelines.

Comment 2: Enrich microbiota section with intestinal permeability (zonulin), Enterococcus gallinarum, microbial metabolites, and therapeutic modulation.
Response: The gut microbiota section has been extensively expanded to include recent findings on intestinal permeability (zonulin), E. gallinarum translocation, microbial metabolites (short-chain fatty acids and tryptophan pathways), and host–microbe immune crosstalk, supported by the most updated literature.

Comment 3: Update emerging-therapy discussion distinguishing approved and investigational agents.
Response: The main focus of this review is on the pathophysiology and immunopathogenic mechanisms of SLE. Therapeutics is out of the scope of this manuscript. We also deleted the “targeted intervention” part in the abstract.

Comment 4: Include sex–gender considerations (X chromosome, TLR7).
Response: A new subsection under Genetic Predisposition elaborates on X-chromosome inactivation escape (TLR7, TASL, CD40L, BTK, XIST, SPEN) and its contribution to female predominance in SLE.

Comment 5: Add a bridging paragraph connecting IFN signatures with B-cell activation.
Response: We now explained how IFN-I signaling promotes extrafollicular B-cell activation and expansion of the autoantibody repertoire, thereby bridging innate and adaptive immunity.

Comment 6: Conclude microbiota section by distinguishing association vs. causation.
Response: We now conclude with the following statement: “Although current findings primarily establish strong associations and biological plausibility rather than definitive causality, their reproducibility across species underscores the gut–immune axis as a pivotal component of lupus pathogenesis.”

Comment 7: Correct minor editorial issues (“INF-1 → IFN-I”; “shading lights → shedding light”).
Response: Corrected throughout the manuscript.

Comment 8: Ensure consistent abbreviation definitions (e.g., Tfh, DN2 B cells, APRIL).
Response: All abbreviations are defined at the first place where they appear in the text.

Reviewer 3 Report

Comments and Suggestions for Authors

1. This review provides a summary of recent advances in the pathophysiology of SLE.
The author should stat the strategy of selection paper including the published year, key words, inclusion and exclusion criteria.
2.Some studies were performed using animal model and some studies were performed using samples from SLE patients. Please clarified it.
3. List of the recently published important original articles in each domain including  innate and Adaptive Immunity, microbiome etc... is required.
4. In the abstract " 
 This review highlights current understanding of SLE immunopathogenesis, the interplay between innate and adaptive immunity, and future directions for targeted interventions to improve patient outcomes." However, few content mention about targeted interventions or improve patient outcomes. The authors might need to revised the statement in the abstract and the main text.
5. In line 260-273, the author briefly introduced the role of epigenetic machanism in the pathogenesis of lupus. I suggest  mention related studies in the innate immune dysregulation or adaptive immune systems section instead.
6. Ref 155 and 156 is the same. Please revised it.

Author Response

Comment 1: Include the strategy of paper selection (year, keywords, inclusion/exclusion criteria).
Response: A methodological note has been added after the Introduction describing the databases (PubMed/MEDLINE, Google Scholar), search window (to January 2025), key terms, and inclusion criteria, following SANRA guidelines.

Comment 2: Clarify which data come from animal models versus human studies.
Response: This point has been briefly addressed in the Introduction and further clarified throughout the manuscript. Instances where animal (murine) models were used have been explicitly identified, while most cited studies are based on human data to ensure clinical relevance.

Comment 3: Provide representative recent original studies in each domain.
Response: Each major section now concludes with representative and up-to-date references, incorporating key findings from both human and murine studies.

Comment 4: Abstract mentions “targeted interventions,” but few therapeutic details are provided.
Response: The abstract now emphasizes mechanistic understanding rather than therapeutic interventions. We deleted “targeted interventions” in the abstract.

Comment 5: Place epigenetic mechanisms within innate/adaptive immunity rather than under genetics.
Response: The epigenetic discussion has been integrated into the Innate Immune Dysregulation section, linking DNA methylation, histone modification, and miRNA regulation directly to IFN signaling and immune activation.

Comment 6: Remove duplicate references (Refs 155 and 156).
Response: Redundant citations have been consolidated and numbering updated.

Round 2

Reviewer 3 Report

Comments and Suggestions for Authors

It is a nice paper.